# Comparison of Electrostatic Spray Drying, Spray Drying, and Freeze Drying for *Lacticaseibacillus rhamnosus GG* Dehydration

**DOI:** 10.3390/foods12163117

**Published:** 2023-08-19

**Authors:** Preethi Jayaprakash, Claire Gaiani, Jean-Maxime Edorh, Frédéric Borges, Elodie Beaupeux, Audrey Maudhuit, Stéphane Desobry

**Affiliations:** 1Laboratoire d’Ingénierie des Biomolécules (LIBio), ENSAIA-Université de Lorraine, 2 Avenue de la Forêt de Haye, BP 20163, 54505 Vandœuvre-lès-Nancy, France; claire.gaiani@univ-lorraine.fr (C.G.);; 2Fluid Air, ZA du Ragon, 28 Rue Louis Pasteur, 44119 Treillières, France; jean-maxime.edorh@spray.com (J.-M.E.); audrey.maudhuit@spray.com (A.M.)

**Keywords:** electrostatic spray drying, spray drying, freeze drying, survivability

## Abstract

Spray drying (SD) is extensively used to encapsulate lactic acid bacteria in large-scale industrial applications; however, bacteria combat several harms that reduce their viability. In this study, a novel technique called electrostatic spray drying (ESD) was used to explore the benefits and disadvantages of using electrostatic charge and lower temperatures in the system. Freeze drying (FD) was used as a reference. The effect of different encapsulation agents, like maltodextrin, arabic gum, and skim milk, on the viability of *Lacticaseibacillus rhamnosus GG* (LGG) was investigated. The initial cell concentration, particle size distribution, aspect ratio, sphericity, scanning-electron-microscopy images, moisture content, water activity, glass transition, rehydration abilities, and survival during storage were compared. Skim milk was proven to be the best protectant for LGG, regardless of the drying process or storage time. A huge reduction in cell numbers (4.49 ± 0.06 log CFU/g) was observed with maltodextrin using SD; meanwhile, it was protected with minimum loss (8.64 ± 0.62 log CFU/g) with ESD. In general, ESD preserved more LGG cells during processing compared to SD, and provided better stability than FD and SD during storage, regardless of the applied voltage. The ESD product analysis demonstrated an efficient LGG preservation, close to FD; therefore, ESD presented to be a promising and scalable substitute for SD and FD.

## 1. Introduction

Due to personal health concerns, consumers expect food to be nutritious and disease-preventative. This is reflected by the popularity of probiotic-based products [1]. Lactic acid bacteria (LAB) are living microorganisms that improve microbial balance in gut flora and have a positive health impact. Food biotechnology relies heavily on lactic acid bacteria to produce a large variety of fermented foods, such as cheese, wine, and sourdough bread [2,3]. Microbial survival depends on the species, strain, and stabilization matrices in the preparation, which will also have an impact on their viability. Microbes must be able to withstand the complex challenges and stresses they face during processing, storage, and transit through the intestine to be in good enough condition to be delivered and to confer the anticipated health benefits [4]. At least 10^8^–10^9^ live bacteria must be present in each gram of probiotic food product [5,6,7]. Bacterial cultures are cultivated in a liquid medium and also prefer subzero transportation and storage temperatures [8]. By adopting encapsulation techniques, such as freeze drying (FD) or spray drying (SD), dry LAB preparations can be produced for incorporation into food. FD is the most common method for drying LAB. This is a costly and time-consuming batch technique. Additional processing procedures, such as grinding, are required to produce microparticles [8,9].

SD protects bacterial cells by entrapping them and creating barriers to adverse environments [10,11,12]. Cells are uniformly distributed throughout the matrix and protected by proteins, lipids, carbohydrates, and/or gums [13]. Several researchers have encapsulated probiotics such as *Lactobacillus rhamnosus GG* [14,15], *Saccharomyces boulardii, Lactobacillus acidophilus, Bifidobacterium bifidum* [16], *Lactobacillus casei* [17], and *Lactobacillus acidophilus* [18] with promising results using various materials, including skim milk, maltodextrin, whey proteins [4], and casein [14]. The drawbacks of using the SD process are oxidative and osmotic stresses and, most notably, high air temperatures (inlet or outlet of the drying system), which affect cell viability [19,20]. These authors [3,17,21] also discovered that compositions that provided the highest protection during SD did not necessarily offer the best storage stability for the dried microorganisms. In terms of industrial aspects, SD provides some advantages over FD such as cost, continuous process, and reproducibility [22].

Electrostatic spray drying (ESD) looks like the conventional SD technique with an electrostatic charge supply. An electrostatic force is established within the droplet and is sustained until dried particles are formed. Technically, under electrostatic force, a solvent (water molecule) typically has the largest dipole moment, whereas a bioactive molecule (such as LAB) has the lowest dipole moment. This enables the feed solution to migrate and transfer the solvent rapidly toward the drying droplet’s outer surface for evaporation. Complete drying occurs easily at lower temperatures (90 °C). Masum et al. [23] encapsulated *Streptococcus thermophilus* and *Lactobacillus delbrueckii* subsp. *Bulgaricus* at a 90 °C inlet temperature with no cell loss; furthermore, this technology has recently been introduced into the market. Only a few studies have validated the encapsulation efficiency of active molecules under the influence of electric voltage [23,24,25]. The selection of protectants is an essential step for the successful drying of living organisms because they improve cell resistance to drying. Unprotected drying harms delicate cell membranes and induces protein denaturation, which can lead to cell death. 

In this study, maltodextrin, skim milk, and arabic gum were used as stabilizing matrices. Previously, these matrices were used to encapsulate several probiotics, including *Lactobacillus plantarum* [26], *Lactobacillus casei* [17], *Bifidobacterium* strains [27], and *Lactobacillus rhamnosus* [1,3,20,28,29]. 

The main objective of the present study was to position ESD as a drying technology for probiotic applications compared to SD and FD, which are well-established processes. In this case, *L. rhamnosus GG* viability was compared immediately after the three drying processes. Electrostatic spray drying was investigated primarily to verify its efficiency on LGG encapsulation and to identify the benefits and limitations for future applications. Additionally, the viability during storage was monitored for twelve weeks to determine the side effects of the processing conditions on the stability. Capsules were analysed using many characterization techniques, such as size, shape, rehydration kinetics, thermal properties, moisture content, and water activity. The various drying processes were compared to obtain a clear perspective on LAB encapsulation strategies for industrial applications. 

## 2. Materials and Methods

### 2.1. Materials

A commercial product, freeze-dried LGG from CHR Hansen (Hørsholm, Denmark), was procured. Maltodextrin (GLUCIDEX^®^ IT 19) was provided by Roquette (Lestrem, France) and Dextrose Equivalent value of 19 was used. Instantgum BB^TM^ (arabic gum) purified from instant acacia gum was purchased from Nexira (Rouen, France). Skim milk powder was provided by Merck (St. Quentin Fallavier, France). Trypton salt broth, MRS (de Man, Rogosa and Sharpe) broth, and agar were purchased from Biokar (Beauvais, France).

### 2.2. Preparation of Feed Solutions

All the feed solutions were prepared at a concentration of 250 g/L. The powders of maltodextrin, skim milk, and arabic gum were rehydrated in distilled water and stirred for 1 h at 550 rpm to prepare the drying feed. The total population of pure LGG (powder) was 11.87 log CFU/g and activated in the distilled water for 30 min at 21 °C at 300 rpm. For bacterial addition, 1.423 g of LGG pure cells were added to 400 mL of matrix solution with 25% (*w*/*w*) solid concentration. The expected number of cells per gram after drying is 9 log CFU/g [30,31]. All feed solutions with LGG were stirred for 30 min at 21 °C to provide good stabilisation until next drying step.

### 2.3. Spray Drying

The SD was performed using a pilot-scale spray dryer (MicraSpray 150; Anhydro, Soeborg, Denmark). The spray-drying conditions were kept constant for all experiments. The drying air rate was set at 86 m^3^/h, inlet temperature at 170 °C, and the atomizing gas pressure was 2 bars. A peristaltic pump was used to deliver feedstock through a bi-fluid nozzle into the spraying chamber at a feed flow rate of 42 g/min. This resulted in an outlet temperature of 85 °C and was maintained throughout the experiments by adjusting the feed rate up and down (38 to 44 mL/min). Similar drying parameters were employed by another author for LGG, especially, the 85 °C outlet temperature is recommended for having limited moisture content in spray-dried microparticles [14]. 

### 2.4. Electrostatic Spray Drying

The ESD was performed using a laboratory-scale electrostatic spray dryer (PolarDry^®^ Model 001, USA). The operating conditions were kept constant for all formulations except the voltage. The drying parameters were as follows: 90 °C inlet temperature; nitrogen flow rate of 25 Nm^3^/h; atomizing gas pressure of 150 kPa; feed rate of 4.5 g/min; electric voltage of 3 kV or 12 kV. The outlet temperature of 42–44 °C was also maintained throughout the experiment [23,32]. The notion of using 3 kV and 12 kV electric voltages is to verify if there are alterations to the end products’ particle size, rehydration, etc., especially to see if the cell viability is influenced under high electric field. To study the impact of the electric voltages on bacteria, electrostatic spray drying of all the formulations was performed in two batches (3 kV and 12 kV).

### 2.5. Freeze Drying

Initially, LGG suspensions were frozen for 24 h at −30 °C and followed by lyophilization in a single-chamber freeze-drier (Christ Model Β 1–6) at a temperature of −30 °C with a chamber pressure of 0.37 mbar. After 48 h of drying, freeze-dried microparticles were produced [31]. The microparticles were crushed to obtain a powder using a simple mortar and pestle to study their particle size and shape.

### 2.6. Storage Conditions

LGG microencapsulation with maltodextrin, skim milk and arabic gum was achieved using the drying processes (SD, ESD, and FD). Immediately after drying, to provide a low water activity (a_w_) and avoid degradation, all the dried microparticles were stored in polyethylene bags at 21 °C (a_w_ close to 0 using P_2_O_5_ salt). 

### 2.7. Characterization Techniques

#### 2.7.1. Particle Size Distribution

Particle size distribution was analysed using laser diffraction granulometry (Mastersizer 3000, Malvern Instruments, Malvern, UK) with an Aero S dry powder dispersion unit. All samples were distributed at an air pressure of 1 bar. The feed rate and hopper length values were regulated between 1 and 15% obscuration. The particle size distribution was determined using the Mie theory. The particle size estimator was d_50_, representing 50% of the particles having smaller diameters. The span value as a polydispersity indicator is estimated using the Equation (1).
(1)$$\mathrm{Span}=\frac{d_{90}-{\;d}_{10}}{d_{50}}$$
where d_10_, d_50_, and d_90_ stand for the diameters of 10, 50, and 90% of the sample particles, respectively. All the microparticles were measured in three replicates.

#### 2.7.2. Particle Shape

The QICPIC (Sympatec GmbH, Clausthal Zellerfeld, Germany) image analysis system was analysed to characterize particle shape of microparticles. The particle images were taken using a camera and assessed using PAQXOS 5.0.1 (Sympatec Gmbh, Wolfenbüttel, Germany). A sample of 0.5 g was passed into a hopper (opening = 1.7 mm) and distributed in the system using compressed air. The aspect ratio (AR) and sphericity (S) of particles were calculated as shown in Equations (2) and (3), respectively. Three repetitions were made for each sample.

AR (0 < AR ≤ 1) represents the elongation and the irregularity range of particles.
AR = d_Feret min_ (µm)/d_Feret max_ (µm)(2)
where d_Feret_ is the Feret diameter, the distance between two parallel tangents of the particle at an arbitrary angle.

The S value (0 to 1). The smaller the value, the higher the irregularity of the particles.
S = P_EQPC_ (µm)/P_reel_ (µm)(3)
where P_EQPC_ is the real perimeter of a circle with the same area; EQPC is the equal projection area of circle; P_reel_ is the real perimeter of the particles.

#### 2.7.3. Scanning Electron Microscopy (SEM)

Microparticles were studied using scanning electron microscopy (JEOL JCM-6000 Plus, Japan). The powders were placed on SEM stubs using double-sided carbon adhesive tape. A sputter coating of gold was applied to the microparticles for 90 s, the equipment was operated at 15 kV, and a distance of 10 mm was used to visualize the morphology of the particles. ESD and SD microparticles were analysed at the magnification of 4000× and FD microparticles at 400×.

#### 2.7.4. Moisture Content (MC)

Moisture content (%) was determined after drying. Initially, 2 g of microparticles were weighed and kept at 103 °C (using drying oven) for 3 h. The variation in dry mass (g) indicates the amount of MC in the microparticles. Three repetitions were performed to obtain an average value.

#### 2.7.5. Water Activity (a_w_) 

The a_w_ of the microparticles (approximately 5 g) was measured using a Hygro palm HP23-AW instrument (Rotronic, Croissy-Beaubourg, France). The microparticles were taken in a propylene container coupled to a digital probe for the a_w_ measurements. Each sample was analysed in triplicate. 

#### 2.7.6. Glass Transition Temperature (Tg) Measurements

Differential Scanning Calorimetry (DSC 2500 discovery series from TA Instruments, Hüllhorst, Germany) was used to determine Tg. Before analysis, all microparticles were adjusted to reach a low a_w_ (close to 0) after storage in the presence of phosphorus pentoxide (P_2_0_5_) for 2 days. Equilibrated microparticles (10–15 mg) were sealed in aluminium Tzero pans (TA Instruments, Hüllhorst, Germany). An empty pan was kept as the reference. A first heating ramp up to 200 °C (10 °C/min) was performed to avoid the thermal history of the sample. Pans were allowed to cool to the initial temperature of −20 °C (10 °C/min). A second heating ramp was performed up to 200 °C (10 °C/min). Tg was determined with the TRIOS analysis software (TA Instruments, Hüllhorst, Germany). All measurements were performed in triplicate.

#### 2.7.7. Reconstitution Time

The reconstitution time of all microparticles in distilled water was measured using a conductimeter (SevenCompact S230, Mettler Toledo, Greifensee, Switzerland). The conductimeter was calibrated with a KCl solution at 1413 μS/cm. The experimental setup consisted of a jacketed glass vessel (250 mL) linked to a thermostat water bath (Eco Gold, Lauda, Germany) to maintain a temperature of 25 °C. The microparticles (2.5 g) were poured all at once after 10 s, and the conductivity was measured every 10 s until total reconstitution. During entire reconstitution time, conductivity changed from an initial value close to 0 μS/cm to a final and maximal value that depends on the powder attributes. The conductivity values were normalized according to Equation (4) [33,34,35].
(4)$$c\left(t\right)=\frac{\kappa \left(t\right)-\kappa_{\mathrm{ini}}}{\kappa_{\mathrm{fin}}-\kappa_{\mathrm{ini}}}$$
c(t): the normalized conductivity (-);κ(t): the conductivity at time t (µS/cm);κ_ini_ (µS/cm): the initial conductivity;κ_fin_ (µS/cm): the final conductivity.


#### 2.7.8. Cell Viability after the Drying Processes

To analyse the effects of drying processes, microbial counts were measured as colony-forming units for 1 g of the solid sample (CFU/g) using the standard plate counting method. Accordingly, 1 g of microparticles was diluted in 10 mL of tryptone salt broth. After tenfold serial dilutions, 1 mL of diluted sample was transferred to MRS agar medium in duplicate. The plates were then incubated at 37 °C for 48 h, and the colony forming units (CFU) per plate were enumerated. Each experiment was performed in triplicate for each formulation.

#### 2.7.9. Cell Viability during Storage

Microparticles of dried powders were analysed for viability at predefined time intervals. Initial survival rate immediately after drying was compared to their viability after 4, 8, and 12 weeks. For each matrix from all drying processes, the same protocol (Section 2.7.8) was followed to enumerate the number of cells per gram. The encapsulated LGG microparticles were sealed in polyethylene bags and stored at 21 °C (a_w_ close to 0) during the entire storage time. All enumerations were performed in triplicate.

#### 2.7.10. Statistical Analysis

All experiments were performed in triplicate and presented as an average value with standard deviations. The results were measured using one-way ANOVA. Tukey’s test was used to identify significant differences between the samples (*p* < 0.05). All analyses were performed using XLSTAT 2022.5.1 (Lumivero, Denver, CO, USA).

## 3. Results and Discussion

### 3.1. Microparticle Characterisation

The particle size distribution, d_50_, and span for the drying processes with different matrices are presented in Figure 1 and Table 1, respectively. Size and shape characteristics are significantly affected by the drying process used to produce the microparticles. First, as shown in Figure 1a, the SD particles exhibited unimodal distribution. Table 1 shows the d_50_ values between 10 and 15 µm for all SD particles. The span value for SD microparticles was the lowest (2.2 and 3.8) compared to other drying techniques because of the uniform size distribution of particles (Figure 1a). In most cases, the SD produces a reliable uniform particle distribution and d_50_ values between 10 and 70 µm [3,36,37]. Secondly, for ESD (Figure 1b,c), the particle size was influenced by the drying parameters. At 3 kV (Figure 1b), two peaks are observed for all matrices; the maximum density peak was 10 µm and the other peak was 100 µm. The same profiles were observed for maltodextrin, skim milk, and arabic gum. ESD particles have d_50_ values between 7 and 12 µm, which are much smaller than those of the other two drying techniques. A minimum span value of 2.2 was observed for ESD but the span value at 12 kV was larger due to the heterogeneous nature (polydisperse) of the microparticles. Similarly, ESD produced microparticles of a minimum particle size of 2–5 µm for drying protein formulations with 5 kV [25]. Nevertheless, the particle distribution results could be also acquired by changing the process parameters, including the nozzle diameter, flow rate, and atomizing pressure [24,38]. FD particle size was much larger than that of SD and ESD. It is evident that the manual crushing performed after the process to convert the original cake-like end products into a powder form has a significant impact on the size and span of FD microparticles. All the microparticles of FD showed a particle size of around 250–325 µm, which was larger than the other two drying techniques.

Considering the particle shape measurements (aspect ratio and sphericity) at d_50_, a higher value indicates a good spherical shape. In Table 1, the SD and ESD particles both have sphericity values of approximately 0.8, which represents the actual spherical shape that could be correlated with the aspect ratio results, with a maximum value of 0.6. FD particles, as expected, are not of spherical shape and so the aspect ratio and sphericity values are below 0.6. Similar irregularities of the particles were observed in correlation with image analysis in previous studies [39].

### 3.2. SEM Images of the Microparticles

Figure 2 shows images of the microparticles with three different matrices produced using the three processes. Both ESD and SD microparticles appeared to have similar morphologies (particle size and shape), and all microparticles were spherical; however, under a microscope, it was evident that they had several protrusions or wrinkles on the surface. In particular, the SD microparticles agglomerated or clustered. Similar morphologies have been observed for maltodextrin, skim milk, and arabic gum while using SD [30,40,41,42]. Particles were partially spherical-shaped. Some of them also had concavities, as often observed in spray-dried microparticles. SD allows the particles to inflate, forming a crust, and in some circumstances, collapse due to rapid evaporation of water and high internal pressure. 

Powder morphologies are influenced by the drying parameters such as drying temperature, feed rate, and composition. ESD parameters with a lower drying temperature of 80 °C conversely caused the microparticles to experience deformation and shrinkage due to slow water diffusion. Differences in voltage parameters (3 kV and 12 kV) did not cause many changes in the physical morphologies of the microparticles. As shown in Figure 2, FD microparticles were visibly irregular in shape or resembled flakes of a much larger size [15,31,31,43]. Overall, in all the drying processes, the microparticles did not show visual cracks or fractures on their surfaces, confirming the good protection of microorganisms with high encapsulation efficiencies. This result could indicate low gas permeability and a long shelf-life period of the microparticles.

### 3.3. MC, a_w_, and Tg Results

MC and a_w_ of all the microparticles are presented in Table 1. The drying parameters, especially the inlet and outlet temperatures of SD and ESD, greatly influenced the residual moisture content values of the microparticles. Aw values played an essential role in preserving powder stability and LGG viability. The particles produced with maltodextrin and skim milk exhibited the lowest moisture content—notably with SD and FD—compared to those with arabic gum. aw below 0.6 is safer for avoiding the microbial growth rate. Microparticles with an aw value of approximately 0.3 are preferred to avoid the Maillard reaction, microbial activity, or oxidation during long storage times [44]. All aw values were measured before storage under controlled temperature. FD produced microparticles with the lowest MC (<2%) and aw (<0.05) compared to ESD and SD. The vacuum used during FD caused moisture vaporization and resulted in a much lower MC and aw compared to other drying techniques. This is consistent with the results of previous studies [15,31]. On the other hand, ESD microparticles had higher MC values at 3 kV than at 12 kV. This might be due to faster water evaporation because of higher electrostatic forces (12 kV electric voltage) developed around the droplet surface during atomization [23,25]. The aw values for both voltages were almost the same, except for the formulation with skim milk at 12 kV. Finally, the SD microparticles had the highest MC values for the three matrices tested, especially arabic gum [30]. 

Conventional spray drying used an 85 °C outlet temperature to encapsulate LGG [14,45]. Comparing SD and ESD, the minimum outlet temperature (42 °C) was achieved with ESD. This low outlet temperature did not affect the MC and a_w_ values of any ESD microparticles for both the 3 kV and 12 kV parameters. The drying efficiency of ESD with low inlet and outlet temperatures was similar to SD with high inlet and outlet temperatures. Both processes showed MC values between 2% and 4%, required for a long shelf-life in the glassy state [44]. 

DSC experiments were performed to determine whether the microparticles were in a glassy state after the three processes. Table 1 shows the Tg data for all the microparticles. Each matrix had a similar Tg (°C) after the three drying processes. Among all the drying processes, maltodextrin microparticles showed a Tg of 100–105 °C, skim milk between 55 and 62 °C, and arabic gum at 48–51 °C. These results confirmed that all microparticles were in a glassy state when stored at 21 °C. The glassy state of microparticles is highly recommended as an effective protecting barrier with low molecular mobility. For bacterial protection, when the storage temperature exceeds Tg (°C), molecular mobility is effective and allows the deterioration of bacterial membranes by lipid oxidation [46], leading to cell death [47]. High Tg was required in all cases to support a stable amorphous structure of microparticles under storage conditions.

### 3.4. Reconstitution Time

The microparticle reconstitution time was performed to verify the capacity of the microparticles to rehydrate in distilled water [33]. Figure 3a shows the instance of reconstitution kinetics of ESD maltodextrin microparticles in distilled water. Figure 3b shows an example of the calculation of the 85% reconstitution time (i.e., 95 ± 7 s). Therefore, the reconstitution times provided in Table 1 represent 85% of the normalized conductivity values. For all matrices and processes, microparticles were rehydrated within less than 500 s. FD microparticles rehydrated faster than the microparticles obtained with the other two drying techniques, due to the highly porous structures developed during water sublimation. ESD microparticles rehydrated faster than microparticles obtained using SD for all matrices. However, the difference in the voltage parameter also influenced the rehydration time based on the matrix type. For instance, microparticles rehydrated faster with powders produced using 12 kV for maltodextrin. In the case of skim milk and arabic gum, the rehydration times were delayed at 12 kV. Reconstitution times for SD microparticles were delayed, especially for arabic gum (485 ± 77 s); in addition, arabic gum, from all the drying processes, required a longer reconstitution time compared to maltodextrin and skim milk matrices.

### 3.5. Cell Viability after the Drying Processes 

The cell content in the suspension before drying was 9 log CFU/g of dry matter of the formulations. Maltodextrin resulted in a significant reduction in viability of approximately 5 logs using the SD process (Figure 4a). This loss of viability was likely due to the harsh drying environments such as inlet/outlet temperatures (170/85 °C, respectively), and/or the disruption of cell membrane integrity during dehydration. Some authors suggest that the bacterial cells’ hydration allows water molecules to interact with the polar groups of the bacterial phospholipid membranes to keep the cells intact. During drying, the water molecules are removed, and membrane integrity is lost. Rehydration causes the cell membrane to disrupt and eventually die. This concept is called the water replacement theory. The loss of cell viability was high with maltodextrin DE 19, causing steric hindrance in the bacterial cell wall and membrane components. Dehydration would not be possible to form sufficient hydrogen bonds with polar residues and replace water molecules; therefore, the bacterial cells would not have been stabilized and died [8,48,49]. 

Among the drying processes, the highest number of cell counts was observed in FD, and this was consistent with other studies when the bacterial cells were encapsulated with maltodextrin [50,51]. Several studies showed that the survivability of LGG is maximum at lower inlet temperatures while optimising the spray drying conditions; however, the reduction in inlet temperature led to an increase in the powders’ moisture content and, subsequently, led to the loss of stability during storage [52]. In this study, ESD achieved drying at a 90 °C inlet temperature with a lower MC (below 4%) and a longer cell-survival rate was observed for 12 weeks. Comparing the results of 3 kV and 12 kV ESD, it is evident that voltage did not disrupt cell integrity and the presence of electrical charges increased the efficiency of drying at a lower temperature of 90 °C compared to 170 °C in SD. 

A maximum of 8.6 log CFU/g was observed after the ESD. During FD, maltodextrin could sustain a survival of 8.4 log CFU/g and the dehydration process did not cause significant losses.

Skim milk matrices provided the best protection with the maximum cell count after FD and ESD processes, 9.4 log CFU/g and 9.02 log CFU/g, respectively (Figure 4b). In the case of SD, there was a minor decrease in cell numbers (8.6 log CFU/g). In previous studies, skim milk was largely used as a protectant for LAB with the spray drying process and produced maximal survival, such as *Lactobacillus acidophilus* La-5 [53]. 

For arabic gum (Figure 4c), the maximum survival of cells was observed with FD, SD, and ESD (12 kV) processes. These results were very close to the initial number of cells. Using ESD at 3 kV, with arabic gum as the encapsulating material, resulted in a minimal reduction of 1.5 log CFU/g, showing an effect of the charges applied. To the best of our knowledge, arabic gum has been used only in combination with saccharides as protectants for LAB and has been effective in sustaining LAB survival after dehydration [50,54]. On comparing these results, with all the matrices and drying processes, all formulations with 25% solid concentration succeeded in protecting LGG, except the maltodextrin formulation during SD.

### 3.6. Cell Viability during 12 Weeks of Storage Time

The decrease in cell count was predominant under adverse storage conditions; consequently, water activity and storage temperature play important roles in cell viability in dried microparticles [18,55]. The samples were stored at 21 °C (a_w_ close to 0) and LGG viability was estimated for 12 weeks.

In Figure 4, there was a maximum reduction in cell numbers during storage in FD microparticles, despite the lower moisture content in dried microparticles. In Figure 4a, maltodextrin formulation decreased from 8.14 log CFU/g to 4.33 log CFU/g in 12 weeks at 21 °C. A similar decrease in cell count was observed when *L. plantarum* was freeze-dried with maltodextrin, especially when stored at room temperature, whereas the maximum number of cells survived when stored at 4 °C [56]. In this case, a minimal a_w_ of 0.1 was achieved; moreover, it was confirmed that the powders were stored in a glassy state at 25 °C. Therefore, the decrease in cell survival during storage could be mainly due to the porous structure of the particles, which would lead to cell oxidation [57]. Although LGG viability immediately after FD was observed to be at maximum, its stability over time was not satisfactory. In general, it is essential that microorganisms are protected not only during the drying process but also during storage [8]. Skim milk was highly efficient in protecting cells from death, with only a slight decrease of 0.75 log CFU/g for over 12 weeks. Arabic gum decreased the number of cells by 5.2 log after 12 weeks of storage. 

In the case of SD, although the initial concentration of cells was drastically reduced because of the SD parameters, the survival of retained cells during storage was satisfactory with a limited log reduction of 1.5 log CFU/g. A similar log reduction was observed for the skim milk microparticles. In previous studies, skim milk has also been shown to be efficient as a bacterial protectant under SD conditions such as LGG [58], Lactobacillus casei LK-1 [17], and Lactobacillus acidophilus [53] during storage. These positive effects of protection have been credited to the calcium and proteins, rather than lactose in the skim milk constituents when rapidly increasing droplet drying temperatures [59]. A drastic reduction in log cells with arabic gum (2.1 log CFU/g) was observed by the end of the 12th week of storage. High moisture content and a_w_ in storage could have triggered the chemical changes in the arabic gum composition and caused LGG cell death [3,60]; however, ESD and FD maintained the highest number of cells with the characteristics of arabic gum during storage. 

Finally, ESD showed the minimum number of dead cells with skim milk and arabic gum after 12 weeks of storage. In fact, ESD Maltodextrin at 3 kV caused approximately 2 log CFU/g loss compared to 12 kV at 1 log CFU/g; meanwhile, skim milk and arabic gum ESD at 12 kV preserved a high number of cells compared to the microparticles produced using 3 kV. Therefore, it was observed that the microparticles produced at 12 kV voltage appeared to protect LGG cells more than the microparticles produced at 3 kV. Dima et al. [61] worked on *Bifidobacterium animalis* subsp. *lactis* under positive or negative charge and studied the cell organisation with maltodextrin based on the difference in polarity. A higher voltage allows high polarity and is inclined to organize the cells at the core of the microparticles [61]. This promoted the stability of probiotic cells under storage, and similarly, microparticles produced at 12 kV had better stability than the low-charged particles.

Significant differences in powder characteristics were observed owing to the voltage connected to the drying system (3 and 12 kV). A voltage supply of 3 kV is sufficient to produce end products with low moisture content, good reconstitution, and high survivability during processing and storage. Using ESD, all matrices managed to protect LGG during storage compared with the other two drying techniques. A lower outlet temperature (44 °C) benefits more sensitive and fragile microorganisms that cannot tolerate heat. After considering skim milk as the best protectant, it was noticed that maltodextrin and arabic gum with the ESD operating conditions preserved the maximum (7–7.3 log CFU/g) for a period of 12 weeks. The physical and chemical characteristics of maltodextrin and arabic gum after ESD likely contributed to the stabilization of cell survivability. These findings support the use of voltages during atomization, without disturbing cell viability during processing or storage.

In FD and SD processes, the survival counts of LGG cells with maltodextrin and arabic gum decreased over time. Dense capsule walls were required to inhibit the penetration of oxygen and extend the integrity of the microparticles at 21 °C. Therefore, the selection of a matrix with appropriate characteristics and the parameters of the drying process were crucial for maintaining the cell viability for long storage periods. 

## 4. Conclusions

Cell viability analysis showed that maximum protection of the cells during the process or storage was observed using skim milk as a protectant. Maltodextrin DE 19 is not a good formulation for use in SD due to the huge loss in cell count (up to 4.2 log CFU/g) during the drying process owing to heat stress (high inlet/outlet temperatures of 170/85 °C). For arabic gum formulation, even if the viability was at maximum after FD and SD processes, their stability over time was insignificant; a loss of 3 log CFU/g and 6.5 log CFU/g was observed after 12 months for FD and SD, respectively. In contrast, ESD arabic gum showed the maximum protection of LAB cells, which were viable at the maximum length of storage. 

In general, the efficiency of electric charges during ESD was demonstrated to evaporate water molecules at a low inlet temperature (90 °C). From the overall results, ESD preserved LGG cells during processing compared to SD and provided better stability than FD and SD during storage, regardless of the applied voltage; additionally, ESD exhibited good cell survivability due to the low porosity of the primary particles and the moderate outlet temperature utilized during the drying process (44 °C). Meanwhile, a_w_ and MC% values were also in accordance with the literature for encapsulating LABs. Therefore, ESD technology is recommended for lactic acid bacteria compared to SD as it eliminates the extreme heat processing conditions; in addition, since the viability results of ESD were close to FD, ESD would be a good alternative to batch process like FD—which is always considered to be efficient but performed for a longer time from an industrial perspective. For further studies, the level of electric charges should be further explored to fully understand its influence on the drying and immobilization of probiotics. Formulations can also be adapted to this encapsulation technique to reach higher viability over time, as shown with skim milk, which is not well accepted at the production scale.

## Figures and Tables

**Figure 1 foods-12-03117-f001:**
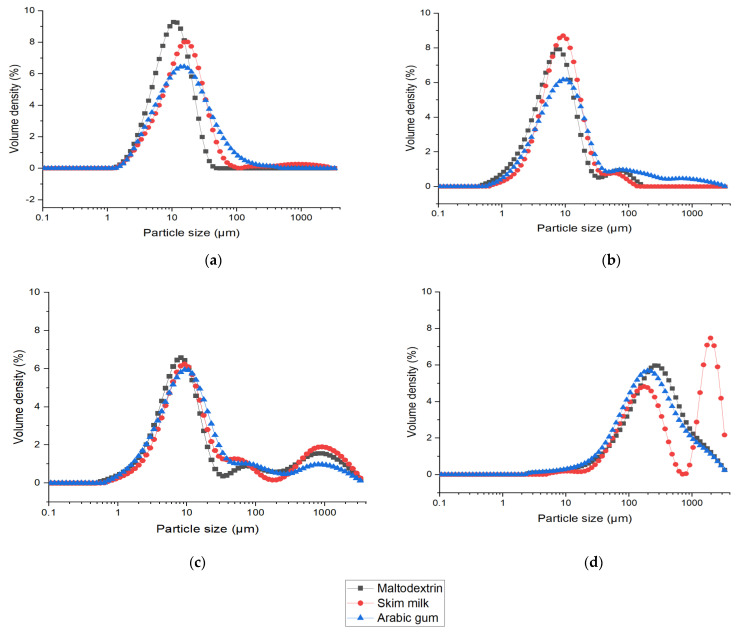
Particle size distribution of SD (**a**), ESD at 3 kV (**b**), ESD at 12 kV (**c**), and FD (**d**) powders.

**Figure 2 foods-12-03117-f002:**
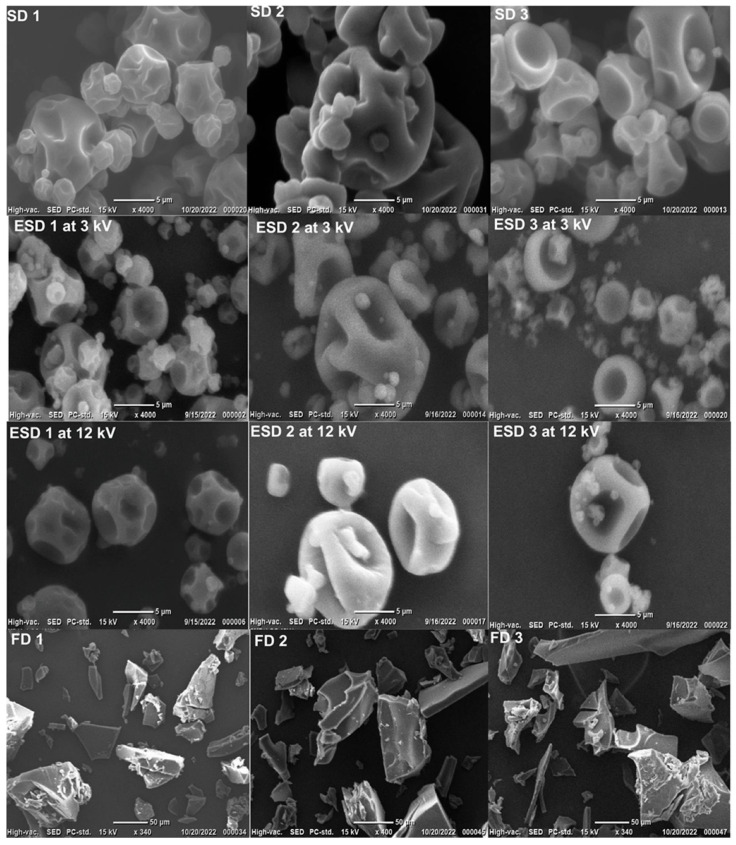
SEM images of ESD, SD, and FD powders with LGG: 1 (Maltodextrin); 2 (Skim milk); and 3 (Arabic gum).

**Figure 3 foods-12-03117-f003:**
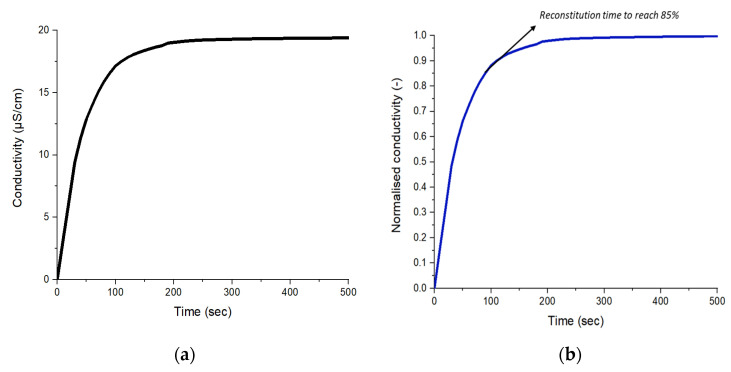
Example of reconstitution kinetics for ESD Maltodextrin LGG microparticles: (**a**) Conductivity raw data (µS/cm) vs. reconstitution time (s); (**b**) Normalized conductivity data vs. reconstitution time (s); black arrow represents the time (s) of 85% reconstitution.

**Figure 4 foods-12-03117-f004:**
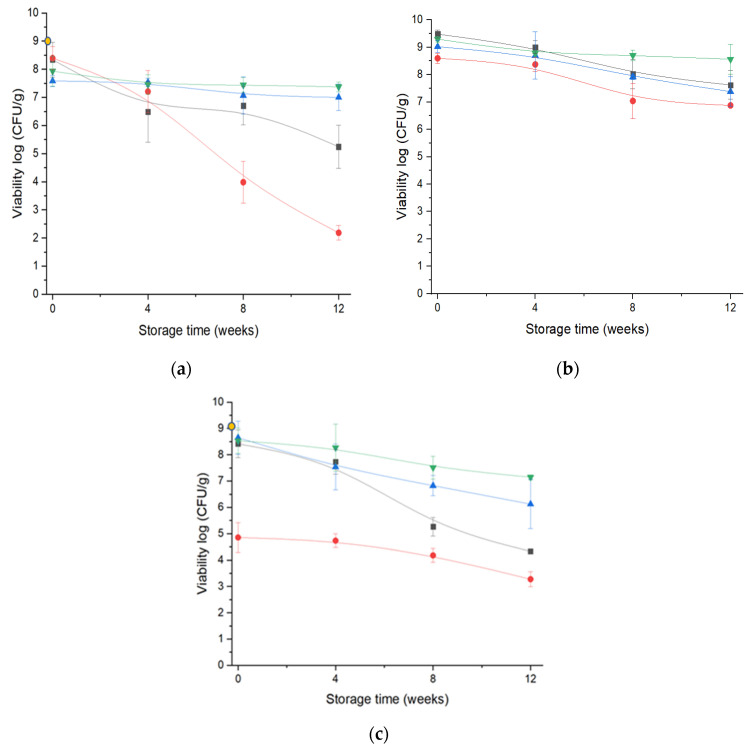
Viability of LGG-encapsulated microparticles, (**a**) Maltodextrin, (**b**) Skim milk, (**c**) Arabic gum, using three drying processes (SD 
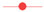
; ESD at 3 kV 
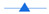
; ESD at 12 kV 
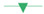
; FD 
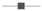
) over storage period of twelve weeks with LGG concentration before drying (
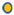
).

**Table 1 foods-12-03117-t001:** Particle size and shape data for all the microparticles: d_50_ (µm), span, aspect ratio, sphericity, a_w_, moisture content (%), reconstitution time (s) to reach 95%, Tg (°C), and cell viability after drying processes (log CFU/g).

Drying Process	Matrices	Particle Size		Particle Shape at d_50_ µm		a_w_	Moisture Content %	Reconstitution Time (s) to Reach 85%	Glass Transition (Tg) °C	Cell Viability after Drying (log CFU/g)
		d_50_ (µm)	Span	Aspect Ratio	Sphericity					
*SD*	*Maltodextrin*	10.1 ± 0.6 ^a^	3.1 ± 0.9 ^a^	0.58 ± 0.02 ^c^	0.87 ± 0.01	0.20 ± 0.01 ^d^	3.90 ± 0.42 ^cd^	110 ± 56 ^ab^	104.31 ± 6.88 ^d^	4.49 ± 0.06 ^a^
	*Skim Milk*	14.3 ± 0.4 ^a^	2.2 ± 0.2 ^a^	0.60 ± 0.01 ^d^	0.75 ± 0.13	0.22 ± 0.03 ^d^	3.83 ± 0.09 ^cd^	285 ± 35 ^de^	62.70 ± 0.55 ^c^	8.57 ± 0.18 ^bc^
	*Arabic gum*	15.4 ± 1.1 ^a^	3.8 ± 0.9 ^a^	0.60 ± 0.01 ^d^	0.82 ± 0.01 ^b^	0.47 ± 0.03 ^e^	9.05 ± 1.48 ^e^	485 ± 77 ^f^	51.15 ± 0.77 ^ab^	8.39 ± 0.40 ^bc^
*ESD at* *3 kV*	*Maltodextrin*	7.1 ± 0.6 ^a^	3.1 ± 0.9 ^a^	0.45 ± 0.01 ^a^	0.90 ± 0.01 ^b^	0.11 ± 0.07 ^c^	3.27 ± 0.29 ^bcd^	95 ± 7 ^ab^	101.90 ± 6.20 ^d^	8.64 ± 0.62 ^bc^
	*Skim Milk*	13.3 ± 7.5 ^a^	2.2 ± 0.1 ^a^	0.61 ± 0.02 ^e^	0.82 ± 0.01 ^b^	0.11 ± 0.05 ^c^	2.97 ± 0.47 ^abcd^	135 ± 7 ^abc^	54.03 ± 5.79 ^abc^	9.02 ± 0.21 ^c^
	*Arabic gum*	10.0 ± 0.1 ^a^	7.6 ± 4.7 ^b^	0.58 ± 0.01 ^c^	0.85 ± 0.04 ^b^	0.10 ± 0.07 ^bc^	4.23 ± 0.54 ^d^	264 ± 35 ^cde^	48.64 ± 0.69 ^ab^	7.58 ± 0.19 ^b^
*ESD at* *12 kV*	*Maltodextrin*	9.8 ± 0.2 ^a^	76.1 ± 3.5 ^d^	0.52 ± 0.01 ^b^	0.88 ± 0.01 ^b^	0.13 ± 0.04 ^c^	2.35 ± 0.58 ^abcd^	62 ± 7 ^ab^	107.06 ± 1.03 ^d^	8.53 ± 0.48 ^bc^
	*Skim Milk*	11.6 ± 1.1 ^a^	70.8 ± 11.6 ^d^	0.63 ± 0.01 ^e^	0.82 ± 0.01 ^b^	0.05 ± 0.01 ^ab^	2.15 ± 0.15 ^abcd^	180 ± 56 ^bcd^	52.08 ± 3.95 ^abc^	9.29 ± 0.10 ^c^
	*Arabic gum*	10.8 ± 0.8 ^a^	21.8 ± 19.8 ^c^	0.66 ± 0.01 ^f^	0.84 ± 0.02 ^b^	0.11 ± 0.01 ^c^	3.98 ± 0.06 ^cd^	362 ± 5 ^ef^	43.89 ± 2.50 ^a^	7.93 ± 0.53 ^bc^
*FD*	*Maltodextrin*	323.0 ± 97.6 ^b^	4.1 ± 0.2 ^a^	0.62 ± 0.01 ^e^	0.70 ± 0.01 ^a^	0.03 ± 0.05 ^a^	0.98 ± 0.06 ^a^	15 ± 1 ^a^	100.81 ± 7.20 ^d^	8.45 ± 0.52 ^bc^
	*Skim Milk*	301.0 ± 7.1 ^b^	5.7 ± 3.1 ^ab^	0.63 ± 0.01 ^e^	0.68 ± 0.01 ^a^	0.04 ± 0.04 ^a^	1.96 ± 0.05 ^abc^	24 ± 4 ^a^	58.85 ± 1.37 ^bc^	9.5 ± 0.14 ^c^
	*Arabic gum*	261.5 ± 78.5 ^b^	3.9 ± 0.6 ^a^	0.62 ± 0.01 ^e^	0.67 ± 0.01 ^a^	0.03 ± 0.04 ^a^	1.50 ± 0.06 ^ab^	104 ± 55 ^bcd^	49.52 ± 3.70 ^ab^	8.34 ± 0.62 ^bc^

Average ± standard deviation with different letters within the same column significantly according to Tukey’s test (*n* = 3, *p* < 0.05).

## Data Availability

Data sharing is not applicable to this article.

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
