# Peer review of "Comparison of Electrostatic Spray Drying, Spray Drying, and Freeze Drying for Lacticaseibacillus rhamnosus GG Dehydration"

_foods, 2023, doi:10.3390/foods12163117_

Round 1

Reviewer 1 Report

This research article compare the L. rhamnosus GG viability after three different drying processes. Meanwhile, the effect of three different encapsulation agents was investigated. The results showed maximum protection of the cells during the process or storage was achieved using skim milk as a protectant and ESD as a technology. This is a very technical articles and important to the food industry, safety and quality improvement. However, several comments need to be improved listed as follows:

1. For Fig. 1, the author uses curves of different colors to represent it. What do these curves represent? Please explain in the figure illustration.

2. About Table 1, variance looks to high for Arabic gum Moisture content like 9.05 ± 1.48. Why is there a relatively large variation here?

3. For Fig. 2, the electron microscope images of freeze-drying method seem to have smaller multiples than other images, why not compare them here under the same multiples?

4. The conclusion looks like a summary, the author need overall conclusion, limitations and future direction.

5. It is advised the authors to cite some recent published articles which have been published in Foods.

6. There are some formatting errors in the article, please further modify according to the requirements of the journal.

Author Response

The authors wish to thank the reviewer for the minor comments, which helped to improve the manuscript quality. We hope that this revision of the manuscript permitted to meet the quality requirements of the journal. Below are the detailed point-by-point answers to reviewer comments. Furthermore, all the minor changes were also marked in red in the revised manuscript.

Reviewer 1

This research article compare the L. rhamnosus GG viability after three different drying processes. Meanwhile, the effect of three different encapsulation agents was investigated. The results showed maximum protection of the cells during the process or storage was achieved using skim milk as a protectant and ESD as a technology. This is a very technical articles and important to the food industry, safety and quality improvement. However, several comments need to be improved listed as follows:

  1. For Fig. 1, the author uses curves of different colors to represent it. What do these curves represent? Please explain in the figure illustration.

Thank you for noticing, I added the figure illustration.

  1. About Table 1, variance looks to high for Arabic gum Moisture content like 9.05 ± 1.48. Why is there a relatively large variation here?

Arabic gum SD, moisture content values were higher, and thus, the drying experiments were repeated and chosen the consistent values to represent. However, the difference must be due to the nature of arabic gum during conventional spray drying process.

  1. For Fig. 2, the electron microscope images of freeze-drying method seem to have smaller multiples than other images, why not compare them here under the same multiples?

Freeze dried powders had much larger particle sizes compared to other techniques (ESD & SD) and in SEM, smaller multiples or scale of SEM, the pictures were complex to have any additional information on the morphologies.

  1. The conclusion looks like a summary, the author need overall conclusion, limitations and future direction.

I concluded the paper as you mentioned, lines 454-468 shows the recently added sentences.

  1. It is advised the authors to cite some recent published articles which have been published in Foods.

Yes, I added recently published articles (reference number 10 to 12) in Foods, related to our study.

  1. There are some formatting errors in the article, please further modify according to the requirements of the journal.

Yes, I did this according to the journal instructions.

Reviewer 2 Report

The title of the manuscript is not clear enough and is a little vague. Please rewrite it in a better way.

The abstract of the article appears to be well-written and concise. However, there are a few minor mistakes that can be addressed:

In the abstract, I cannot see the conclusion part.

Moreover, in the comparison of drying methods, explain it with numbers. For example, viability of bacteria in the final powder, etc.

Introduction

Line 78: Mention the storage time exactly, not as several weeks.

Your literature review in the introduction part is weak.

What is the novelty and importance of this work that should be clearly presented in the introduction?

Materials and methods

All parts of methods need reference.

Activation of LGG freeze-dried powder is missing. Please mention that.

Line 97-98: The temperature of stabilizing solutions must be mentioned.

The materials and methods section is numbered as section 2, and all subsections must be numbered as subcategories of 2. Correct the numbering in your article, specifically in the characterization part.

Line 131: What is 3 characterization techniques? Clarify this.

In section 3.9, mention the packaging material and conditions of encapsulated LGG.

Results and discussion

This part is written in a good way.

Conclusion

The last sentence in the conclusion part is not sufficient and must be clarified and rewritten.

In this part, write the exact application and benefit of the results of this research."

no comment

Author Response

The authors wish to thank the reviewer for the minor comments, which helped to improve the manuscript quality. We hope that this revision of the manuscript permitted to meet the quality requirements of the journal. Below are the detailed point-by-point answers to reviewer comments. Furthermore, all the minor changes were also marked in red in the revised manuscript.

The title of the manuscript is not clear enough and is a little vague. Please rewrite it in a better way.

As the reviewer suggested, the title is re-written as “Comparison of Electrostatic Spray drying, Spray Drying and Freeze Drying for Lacticaseibacillus rhamnosus GG dehydration”

The abstract of the article appears to be well-written and concise. However, there are a few minor mistakes that can be addressed:

In the abstract, I cannot see the conclusion part.

Line 20-25, I added few lines as a conclusion to the study.

Moreover, in the comparison of drying methods, explain it with numbers. For example, viability of bacteria in the final powder, etc.

I modified the values in the lines 20 and 22 accordingly.

Introduction

Line 78: Mention the storage time exactly, not as several weeks.

I changed it.

Your literature review in the introduction part is weak.

What is the novelty and importance of this work that should be clearly presented in the introduction?

I added the information on the lines 78 to 80.

Materials and methods

All parts of methods need reference.

References for all drying methods were mentioned in the line 102, 113, 120 and 129.

Activation of LGG freeze-dried powder is missing. Please mention that.

Line 99 mentioned the activation protocol of LGG.

Line 97-98: The temperature of stabilizing solutions must be mentioned.

Now, it is the line 102 and I added the information.

The materials and methods section is numbered as section 2, and all subsections must be numbered as subcategories of 2. Correct the numbering in your article, specifically in the characterization part.

Yes, I changed the numbering like you suggested.

Line 131: What is 3 characterization techniques? Clarify this.

It is a mistake while aligning the paper and I removed it now.

In section 3.9, mention the packaging material and conditions of encapsulated LGG.

Now, section 2.7.9 and line 217, I mentioned the required information.

Results and discussion

This part is written in a good way.

Conclusion

The last sentence in the conclusion part is not sufficient and must be clarified and rewritten.

In this part, write the exact application and benefit of the results of this research."

I re-wrote the sentences with additional information line 463-471.
